# Proline Oxidation Supports Mitochondrial ATP Production When Complex I Is Inhibited

**DOI:** 10.3390/ijms23095111

**Published:** 2022-05-04

**Authors:** Gergely Pallag, Sara Nazarian, Dora Ravasz, David Bui, Timea Komlódi, Carolina Doerrier, Erich Gnaiger, Thomas N. Seyfried, Christos Chinopoulos

**Affiliations:** 1Department of Biochemistry and Molecular Biology, Semmelweis University, 1094 Budapest, Hungary; pallag.gergely@semmelweis-univ.hu (G.P.); nazarian.sara@med.semmelweis-univ.hu (S.N.); ravasz.dora@med.semmelweis-univ.hu (D.R.); b.d4vid@gmail.com (D.B.); 2Oroboros Instruments, 6020 Innsbruck, Austria; timea.komlodi@oroboros.at (T.K.); carolina.doerrier@oroboros.at (C.D.); erich.gnaiger@oroboros.at (E.G.); 3Biology Department, Boston College, Chestnut Hill, MA 02467, USA; seyfridt@bc.edu

**Keywords:** proline dehydrogenase, substrate-level phosphorylation, coenzyme Q, reducing equivalent

## Abstract

The oxidation of proline to pyrroline-5-carboxylate (P5C) leads to the transfer of electrons to ubiquinone in mitochondria that express proline dehydrogenase (ProDH). This electron transfer supports Complexes CIII and CIV, thus generating the protonmotive force. Further catabolism of P5C forms glutamate, which fuels the citric acid cycle that yields the reducing equivalents that sustain oxidative phosphorylation. However, P5C and glutamate catabolism depend on CI activity due to NAD^+^ requirements. NextGen-O2k (Oroboros Instruments) was used to measure proline oxidation in isolated mitochondria of various mouse tissues. Simultaneous measurements of oxygen consumption, membrane potential, NADH, and the ubiquinone redox state were correlated to ProDH activity and F_1_F_O_-ATPase directionality. Proline catabolism generated a sufficiently high membrane potential that was able to maintain the F_1_F_O_-ATPase operation in the forward mode. This was observed in CI-inhibited mouse liver and kidney mitochondria that exhibited high levels of proline oxidation and ProDH activity. This action was not observed under anoxia or when either CIII or CIV were inhibited. The duroquinone fueling of CIII and CIV partially reproduced the effects of proline. Excess glutamate, however, could not reproduce the proline effect, suggesting that processes upstream of the glutamate conversion from proline were involved. The ProDH inhibitors tetrahydro-2-furoic acid and, to a lesser extent, S-5-oxo-2-tetrahydrofurancarboxylic acid abolished all proline effects. The data show that ProDH-directed proline catabolism could generate sufficient CIII and CIV proton pumping, thus supporting ATP production by the F_1_F_O_-ATPase even under CI inhibition.

## 1. Introduction

Taggart and Krakaur discovered proline oxidation in mitochondria isolated from rabbit kidneys in 1949 [1]. In 1962, Johnson and Strecker reproduced this finding in rat liver mitochondria [2], and in 1986, McKnight and Hird demonstrated proline oxidation in mitochondria from other rat tissues [3]. Although proline oxidation has received the most attention in insect flight muscles [4,5,6], and pioneering studies by Phang and co-workers have established a specialized role for this amino acid in cancer metabolism [7,8].

Hereby, we investigated the effect of proline oxidation in providing sufficient bioenergetic drive for supporting mitochondrial ATP production when respiratory Complex CI, which was inhibited using rotenone. Being mindful that proline catabolism exhibits strong tissue-dependence, we measured the extent of proline oxidation in isolated mitochondria that was obtained from mouse liver, kidney, heart, and brain. Proline catabolism (and in particular ProDH activity) has only been investigated in mouse liver mitochondria to the best of our knowledge [9].

The metabolism of proline in mitochondria is outlined in Figure 1 shown below. For a more detailed review see [10,11]. As shown in the scheme, proline enters mitochondria through a bi-directional transporter. Unlike the plasma membrane, in which twelve transporters have been identified and characterized [12], the transport of proline across the mitochondrial inner membrane has been regarded only as “energy-dependent” [13] and is mediated by two entities: a proline uniporter and a proline/glutamate antiporter [14]. The genetic identities of these transporters remain unknown. The rate of proline transport by these two mitochondrial transporters is similar to the first two steps of proline oxidation, indicating that transport is not a limiting factor for proline metabolism [13]. Once inside the matrix, the proline dehydrogenase/proline oxidase (ProDH) reversibly converts proline to pyrroline-5-carboxylate (P5C). There are two ProDH enzymes: ProDH1, which converts L-proline to P5C, and ProDH2, which catalyzes the conversion of hydroxyproline to pyrroline-3-hydroxy-5-carboxylate [15]. ProDH2 can also use proline as a substrate, but with much lower efficiency [15]. ProDH enzymes are FAD-bound, thereby reducing UQ [9,16]. In 1965, Erecińska identified the requirement of ubiquinone in proline oxidation [17]. Reduced Q (ubiquinol, UQH_2_) fuels CIII, subsequently transferring electrons to CIV—provided that a suitable final electron acceptor is available [9]. P5C non-enzymatically tautomerizes to glutamate semi-aldehyde (GSA). GSA may have two fates: (i) transamination with glutamate to ornithine and oxoglutarate (Og; α-ketoglutarate) by ornithine aminotransferase (OAT), and/or (ii) oxidation to glutamate by delta-1-pyrroline-5-carboxylate dehydrogenase (ALDH4A1) concomitantly reducing NAD(P)^+^ to NAD(P)H. Glutamate may then enter the citric acid cycle either through glutamate dehydrogenase yielding oxoglutarate, or through transamination with oxaloacetate to oxoglutarate and aspartate by aspartate aminotransferase (ASAT).

Being mindful of the above metabolic considerations, we investigated the effects of adding proline to mitochondria by measuring: (i) oxygen consumption (final electron acceptor of CIV that receives electrons from CIII fueled by UQH_2_ generated by ProDH), (ii) NAD^+^ reduction reflecting ALDH4A1, GDH, and citric acid cycle dehydrogenases activities, (iii) ubiquinone reduction by ProDH while converting proline to P5C, (iv) generation of a mitochondrial membrane potential (Δ*Ψ*_mt_) by CIII and CIV proton pump activity, and the subsequent use of these protons by F_1_F_O_-ATPase and downstream events that alter matrix ATP/ADP (in the presence of CI inhibition), and (v) the directionalities of F_1_F_O_-ATPase and the adenine nucleotide translocase (ANT), which are profoundly influenced by the ATP/ADP ratio, [18]—also in the presence of CI inhibition. Data obtained from these experiments were correlated with ProDH activity values that were estimated from the same tissues.

We report that proline could maintain ATP formation by the mitochondrial F_1_F_O_-ATPase under CI inhibition only in tissues with high ProDH activity. This effect was mediated by the reduction of ubiquinone (UQ) fueling Complexes CIII and CIV, which in turn generates a membrane potential that is not due to the oxidation of glutamate and is formed by the catabolism of proline. Our results suggest that proline catabolism can bypass a CI blockade, thus preventing bioenergetic collapse.

## 2. Results

### 2.1. Kinetic Characterization of Proline Dehydrogenase in Mitochondria Isolated from Various Mouse Tissues

To the best of our knowledge, ProDH activity in murine tissues has onlybeen reported from a mouse liver model [9] and various organs of rats [3]. Thus, we first measured ProDH catalytic activity content in isolated mitochondria from mouse liver, kidney, brain, and heart, and also determined the apparent *K*_m_ of mouse liver ProDH for proline. As shown in Figure 1A, liver and kidney mitochondria exhibited much higher ProDH activity than brain and heart mitochondria, as similarly reported for rat tissues in [3]. We further determined the apparent *K*_m_ of ProDH for proline in mouse liver mitochondria and report it to be 3.08 ± 0.48 mM (Figure 1B), which was estimated by non-linear fitting. Values of 2.4 mM for fetal and adult rat liver mitochondria [19] and 0.42–1.2 mM in DLD-POX cells [9] have been previously reported.

### 2.2. Liver and Kidney Mitochondria Respire on Proline

Having measured the tissue-dependent ProDH activity in isolated mitochondria from mouse tissues, we sought to establish the extent of proline catabolism as a respiratory substrate. The effect of proline on oxygen consumption rates (and all subsequent experiments) was tested in the 0.25–10 mM concentration range. This is because the normal human plasma concentration of this amino acid is in the range of 100–250 μM [13,20,21,22,23]. However, in patients suffering from type 2 diabetes, obesity, insulin resistance [24], or cancer-associated cachexia [25], an almost two-fold increase has been reported. Similar to humans, plasma proline concentration in the range of 0.25–0.3 mM has been observed in rats [26].

As shown in Figure 1C for liver and 1H for kidney, proline led to a dose-dependent (0.25–10 mM) increase in LEAK and OXPHOS respiration (kinetically saturating ADP [27]) of isolated mitochondria. The proline-induced increases in LEAK and OXPHOS respiration were, however, masked if glutamate and malate (Figure 1D for liver and 1I in kidney); glutamate, malate, and β-hydroxybutyrate (βOH, Figure 1E in liver); or glutamate, malate, and itaconate (Figure 1J in kidney) were present. βOH increases the NADH/NAD^+^ ratio due to the high activity of β-hydroxybutyrate dehydrogenase in the liver, while itaconate (Itac) limits mitochondrial substrate-level phosphorylation as it is a preferred substrate for succinate-CoA ligase, which also leads to a CoASH-trap [28], thus exerting metabolic pressure on the overall citric acid cycle. The increases in OXPHOS capacities were not merely additive but strongly synergistic to that conferred by succinate in liver (Figure 1F,G) and kidney mitochondria (Figure 1K,L). OXPHOS capacity with the succinate and proline substrate combination (*J*_SPro_) was 1.4- and 1.2-fold higher than the arithmetic sum *J*_Pro_ + *J*_S_ (Table 1), thereby demonstrating a synergistic effect and excess additivity [29]. In contrast, additivity in the LEAK state was partially or completely additive in liver mitochondria. In kidney mitochondria, additivity was even negative in the LEAK state, indicating suppression of LEAK respiration when proline was added to succinate (Table 1; Appendix A).

### 2.3. Effect of Proline on ΔΨmt, NADH Autofluorescence and Q Redox State of Isolated Mitochondria

Being mindful that, in the presence of NADH-linked substrates (glutamate, malate, βOH), the addition of proline did not yield an additional increase in the oxygen consumption rate, we questioned whether this was because of the downstream production of glutamate (see Figure 1) that would fuel the citric acid cycle or be a limitation of the measurement itself—having saturated the capacity of CIV transferring electrons to molecular oxygen. Therefore, we examined the effects of proline on other bioenergetic readouts, namely Δ*Ψ*_mt_, NADH autofluorescence, and the Q redox state of isolated mitochondria. As shown in Figure 2A for liver and Appendix A for kidney, mitochondria were added in the buffer without substrates and allowed to develop a transient membrane potential before exhibiting a depletion of endogenous substrates, thereby leading to a complete loss of Δ*Ψ*_mt_. Subsequently, proline was added in boluses at the concentrations indicated in the panels, thereby leading to progressive polarization. The further addition of glutamate (G) and malate (M) did not yield any further increase in Δ*Ψ*_mt_. On the other hand, the addition of succinate (S; Figure 2B for liver and Appendix A for kidney mitochondria) led to a further decrease in safranine O fluorescence, which is indicative of a gain in Δ*Ψ*_mt—_even in the presence of rotenone. Importantly, the presence of rotenone did not abolish the proline-induced polarization. However, the CIII inhibitor, myxothiazol, completely inhibited the proline-induced polarization in both liver (Figure 2D) and kidney (Appendix A) mitochondria. On the other hand, atpenin A5 blocked the succinate-induced gain in Δ*Ψ*_mt_, but the proline-induced changes remained unaffected (Figure 2C for liver mitochondria).

In congruence with the data showing Δ*Ψ*_mt_, additions of proline to mitochondria led to a dose-dependent increase in NADH autofluorescence; NADH autofluorescence was recorded fluorometrically using two different equipment: a Hitachi F-7000 fluorescence spectrophotometer or an Oroboros NextGen-O2k. Data obtained with the Hitachi are shown in Figure 2E (for liver mitochondria) and Appendix A (for kidney mitochondria), while those obtained with the NextGen-O2k are shown in Figure 2G (for liver mitochondria) and Appendix A (for kidney mitochondria). The oxygen consumption rate, NADH autofluorescence, and rhodamine 123 fluorescence (indicative of Δ*Ψ*_mt_) were simultaneously recorded and shown in Figure 2F–H, respectively, for the liver, and Appendix A for the kidney, which are aligned on the dashed grey lines using the NextGen-O2k. As shown in Figure 2E,G, Appendix A, the subsequent addition of rotenone led to greater increases in NADH autofluorescence depending on the amount of proline added to mitochondria. This additional increase in NADH depending on the proline concentration reflects the NADH originating from the reaction catalyzed by ALDH4A1, which is upstream of glutamate (see Figure 1).

To strengthen the above conclusions that proline is catabolized in liver and kidney mitochondria, we recorded the quinone (Q) redox state using an Oroboros NextGen-O2k, which was measured simultaneously with the oxygen consumption rate and rhodamine 123 fluorescence (indicative of Δ*Ψ*_mt_). As shown in Figure 3A–C (for liver mitochondria) and Appendix A (for kidney mitochondria), bolus additions of proline (concentrations indicated in the panels) led to measurable increases in oxygen consumption rates, Q reduction, and gains in Δ*Ψ*_mt_. The increases in Q reduction were less pronounced in the presence of rotenone (Figure 3E for liver and Appendix A for kidney mitochondria) than in the absence of this CI inhibitor (Figure 3B and Appendix A for liver and kidney mitochondria, respectively). The ‘dampening’ of changes in the Q reduction in the presence of rotenone (which is expected to lead to an increase in matrix NADH/NAD^+^) reflect the part of proline catabolism that is downstream from ProDH, specifically the NAD^+^-requiring ALDH4A1, GDH, and OgDH steps; they probably signal an increase of substrate(s), leading to a moderate decrease in ProDH activity and yielding the less-pronounced effect of proline on the Q redox state in the presence of rotenone. The changes in Q reduction are shown as a non-calibrated signal; thus, at the end of the experiments, succinate was added and was followed by SF6847. This was done in order to provide a semi-quantitative estimation of the Q reduction by proline compared to the reduction by succinate dehydrogenase and the oxidation of the electron transfer system (ETS) due to the uncoupler. As expected, in the presence of CIII inhibitor and myxothiazol, proline led to no changes in the Q reduction state (Figure 3H), which was monitored simultaneously with the oxygen consumption rate (Figure 3G) and rhodamine 123 fluorescence (Figure 3I).

In accordance with the lower ProDH activities measured in brain and heart mitochondria (Figure 1A), the addition of proline to brain (Appendix A) or heart (Appendix A) mitochondria led to a moderate gain of Δ*Ψ*_mt_. Accordingly, while measuring NADH autofluorescence or Q reduction, the addition of proline to brain or heart mitochondria led to evanescent changes—as shown in Appendix A for NADH autofluorescence in brain and heart mitochondria and Appendix A for the Q redox state in brain mitochondria, which was measured simultaneously with the oxygen consumption rate (Appendix A) and rhodamine 123 fluorescence (Appendix A).

### 2.4. Proline Oxidation Is Sensitive to Tetrahydro-2-Furoic Acid (THFA) and S-5-Oxo-2-Tetrahydrofurancarboxylic Acid (S-5-oxo)

Tetrahydro-2-furoic acid (THFA) and S-5-oxo-2-tetrahydrofurancarboxylic acid (S-5-oxo) have been described as specific inhibitors of ProDH, the latter compound branded as a “second-generation” from the former [30,31]. In our hands, 10 mM THFA inhibited the ProDH activity of liver and kidney mitochondria to a great extent (>80 % inhibition, see Figure 4A), while S-5-oxo had little—if any—effect at 5 mM. THFA exhibited a dose-dependent decrease in LEAK and OXPHOS respiration in both liver (Figure 4B) and kidney (Figure 4D) mitochondria. At 10 mM, S-5-oxo only inhibited proline oxidation in liver (Figure 4C), but not kidney (Figure 4E), mitochondria.

### 2.5. Effect of Proline Dehydrogenase Inhibitors on Proline-Induced Changes in Q Redox State and ΔΨmt

Based on the inhibitory effects of THFA and S-5-oxo on ProDH activity and proline-mediated respiration, we sought to establish the effect of these inhibitors on other proline-mediated bioenergetic read-outs. For this, we investigated the effects of the compounds on proline-induced changes in the Q redox state and Δ*Ψ*_mt_. As shown in Figure 4, liver mitochondria THFA and S-5-oxo abolished the proline-induced Q reduction (proline added after ADP and the CI inhibitor rotenone) and decreased rhodamine 123 fluorescence, which is indicative of Δ*Ψ*_mt_. The Q redox state and rhodamine fluorescence were recorded simultaneously with oxygen consumption and were aligned on the dashed grey lines. The results obtained from THFA are shown in Figure 4F–H, and for S-5-oxo they are shown in Figure 4I–K. Various traces imply different substrate combinations and different concentrations of the ProDH inhibitor, which are detailed in the legend. Qualitatively similar results were obtained from kidney mitochondria, which are shown in Appendix A. In Appendix A, the dose-dependent effect of THFA was examined for the proline-induced changes of safranine O fluorescence, which is indicative of Δ*Ψ*_mt_, while in Appendix A, they were recorded simultaneously using a NextGen-O2k (aligned in the grey dashed lines). The effects of THFA on respiration, the Q redox state, and rhodamine 123 fluorescence are shown. Various traces imply different substrate combinations and different concentrations of the ProDH inhibitor, which are detailed in the legend.

### 2.6. Effect of Proline on ANT Directionality

From the above experiments, it is evident that mouse liver and kidney (and to a much lesser extent brain and heart) mitochondria, and in accordance with the corresponding ProDH activities, proline is catabolized and generates Δ*Ψ*_mt_ in a manner that is dependent on CIII (and CIV) function. In order to address if this is sufficient for maintaining matrix ATP levels, we interrogated the directionality of ANT, a parameter that is profoundly sensitive to matrix [ATP]/[ADP] [18,32,33]. ANT directionality was addressed using a biosensor test developed by us, in which the effect of the ANT inhibitor carboxyatractyloside (CAT) is examined for Δ*Ψ*_mt_ during ADP-induced respiration but after targeted inhibition of the ETS [32]. Briefly, the adenine nucleotide exchange through the ANT is electrogenic, since one molecule of ATP^4–^ is exchanged for one molecule of ADP^3–^ [34]. In fully energized mitochondria, the export of ATP in exchange for ADP costs ~25% of the total energy produced [35]. Therefore, during the forward mode of ANT, the abolition of its operation by CAT leads to a gain of Δ*Ψ*_mt_, whereas during the reverse mode of ANT, the abolition of its operation by the inhibitor leads to Δ*Ψ*_mt_ loss.

A generalized scheme is depicted in [36]. As shown in Figure 5, mouse liver mitochondria proline kept the ANT operating in its forward mode irrespective of the substrate combinations used and indicated if it was present before or after ETS inhibition by rotenone (Figure 5A–D).

However, when ETS was blocked at the level of CIII with myxothiazol (Myx, 5E) or CIV with cyanide (CN, 5F,G), proline failed to maintain the ANT in forward mode, implying that any changes conferred by proline required the uninterrupted operation of both CIII and CIV. Importantly, an excess of glutamate could not reproduce the effects of proline (Figure 5D), thereby indicating that the effects of proline are not due to the downstream formation of glutamate. As expected, the effects of proline in liver mitochondrial ANT directionality were sensitive to inhibition by both THFA (Figure 5H,I) and S-5-oxo (Figure 5J). Qualitatively similar results were obtained with kidney (Appendix A), brain (Appendix A), and heart (Appendix A) mitochondria, which were always in accordance with ProDH activities; however, S-5-oxo was much less potent than THFA in kidney mitochondria. In experiments using kidney mitochondria, 2-ketobutyrate was also included; 2-ketobutyrate negates mSLP due to the ATP-consuming propionyl-CoA carboxylase step [37], and this was used as a way to limit mSLP and examine any beneficial effects of proline that could be affected by the ProDH inhibitors. As was also expected, proline had no effect on the ANT directionality of mitochondria subject to anoxia (i.e., complete non-pharmacological CIV inhibition)—as shown in Figure 6. Interestingly, proline exhibited a dose-dependent effect in CAT-induced Q reduction in the presence of glutamate and malate (Figure 7B), which was dampened if βOH was concomitantly present (Figure 7E). These recordings were simultaneously measured with the oxygen consumption rate and rhodamine 123 fluorescence (indicative of Δ*Ψ*_mt_) (Figure 7A,D and Figure 7C,F, respectively) using the NextGen-O2k in isolated liver mitochondria. Qualitatively similar results were obtained with kidney mitochondria (Appendix A). The reason(s) for the CAT-induced changes in the Q redox state during CI inhibition, and as a function of proline, were not investigated further. As expected, when CIII was inhibited by myxothiazol (Figure 8A–C) or under anoxia (Figure 8D–F), these phenomena were not observed. Qualitatively similar results were obtained with anoxic (Appendix A) and CIII-inhibited kidney mitochondria (Appendix A).

### 2.7. Proline Oxidation Maintains F_1_F_O_-ATPase in Forward Mode during Complex I Inhibition

When mitochondria depend on pyruvate or other widely used NADH-linked substrates, such as glutamate or oxoglutarate, to support respiration, inhibition of the electron transport system at the level of CI leads to F_1_F_O_-ATPase reversal [38,39]. This results in the maintenance of a Δ*Ψ*_mt_ value that is no higher than the reversal potential of the F_1_F_O_-ATPase [18,33]. Being mindful that proline fuels mitochondria through ProDH reducing Q reminiscent of CI bypass [40,41,42], we sought to address the directionality of the F_1_F_O_-ATPase in mitochondria supported by proline in the presence of the CI inhibitor rotenone. This was done in a manner similar to interrogating the ANT directionality but using the F_1_F_O_-ATPase inhibitor oligomycin (Omy) instead of CAT. As shown in Figure 9 for liver and Appendix A for kidney mitochondria, proline added in various concentrations, as indicated in the legends, maintained F_1_F_O_-ATPase in the forward mode; this was not observed if CIII was inhibited by myxothiazol (Myx, 9E, S12F) or if CIV was inhibited by cyanide (CN, 9F, S12G).

In agreement with the data above, THFA (9G, S12C, S12D) could abolish the proline effect. Importantly, the effect of proline maintaining F_1_F_O_-ATPase operation in the forward mode was primarily due to a small but sufficient gain in Δ*Ψ*_mt_, which effectively crossed the reversal potential of the F_1_F_O_-ATPase “to the left”—see [36]. This was deduced from the experiments shown in Figure 9D for liver and Appendix A for kidney mitochondria. In these experiments, the uncoupler SF6847 was titrated to clamp Δ*Ψ*_mt_ to a level that was equal to just before proline addition. Indeed, at these SF6847 concentrations, the effect of proline was abolished. From this, we concluded that proline was maintaining F_1_F_O_-ATPase in the forward mode exclusively because of a gain in Δ*Ψ*_mt_ due to proton pumping by CIII and CIV, which is supported by the Q reduction through ProDH.

Finally, proline exerted an effect on the oligomycin-induced Q reduction in the presence of glutamate and malate (Appendix A), just like CAT (Figure 7B). These recordings were simultaneously made with the oxygen consumption rate and rhodamine 123 fluorescence (indicative of Δ*Ψ*_mt_) (Appendix A, respectively) using the NextGen-O2k. The reason(s) for the oligomycin-induced changes in the Q redox state during CI inhibition as a function of proline were not investigated further.

### 2.8. Fueling Complex III with Duroquinone Only Partially Mimics the Benefits of Proline

As shown from the results above, proline is oxidized by ProDH reducing Q, which is in turn oxidized by CIII and also requires CIV and oxygen as a final electron acceptor. Thus, we sought to compare the effects of proline with duroquinone (DQ), an artificial substrate of CIII. As shown in Figure 10A (for liver) and 10B (for kidney), DQ could partially mimic the effect of proline in maintaining F_1_F_O_-ATPase in the forward mode. Higher concentrations of DQ had deleterious effects, as they probably damage the mitochondrial inner membrane. Thus, proline is a far superior substrate for fueling CIII—though indirectly through Q. Furthermore, the effects of proline could not be reproduced by ornithine (Figure 10C), a metabolite forming GSA by transamination with Og (see Figure 1); being mindful that GSA is downstream to proline oxidation by ProDH, it is concluded that proline effects are exclusively due to upstream Q reduction. In addition, proline could also partially rescue ANT and F_1_F_O_-ATPase operation reversal induced by arsenite (NaAsO_2_)—an inhibitor of dehydrogenases including oxoglutarate dehydrogenase (Figure 10D)—thus its effects are unrelated to the dehydrogenase. Finally, the effects of proline were unaffected by dicoumarol, thus they are not mediated through diaphorases (Figure 10E).

## 3. Discussion

It is a textbook definition that the F_1_F_O_-ATPase reverses, pumping protons out of the matrix even at the cost of ATP consumption, thereby preserving the mitochondrial membrane potential when CI is inhibited, and only NADH-linked substrates are available [29,38,39]. The most important conclusion of the present study is that the above statement does not apply when mitochondria oxidize proline through ProDH. We showed that, in CI-inhibited mitochondria exhibiting a sufficiently high ProDH activity, the reduction of ubiquinone fueling CIII and CIV, which leads to proton pumping, supports Δ*Ψ*_mt_ to a level at which the F_1_F_O_-ATPase maintains ATP production. This finding agrees with a modelling study in which the authors claimed that CI deficiency was maybe compensated for by proline oxidation [43].

Notably, this property of proline is not due to catabolism towards glutamate and oxoglutarate, as further catabolism of either metabolite requires CI activity. However, the effect of proline on the Q redox state was partially sensitive to rotenone, implying the continuity of the pathway Pro -> P5C -> GSA -> G -> Og, where the first step generates reduced Q, while the latter steps require NAD^+^. The ability of proline to maintain the F_1_F_O_-ATPase in the forward mode is genuinely due to the metabolism of proline through ProDH. This leads to Q reduction and the fueling of CIII. Relevant to this, duroquinone (DQ) could partially mimic the effects of proline by directly fueling CIII, but not to the same effect—as DQ is expected to exert undesirable effects such as ROS formation [44]. Several electron transfer pathways converge at the Q-junction, particularly the NADH- and succinate-linked (CI- and CII-linked) pathways, and include the glycerophosphate pathway and fatty acid oxidation pathway with electron entry through the electron-transferring flavoprotein Complex into Q as well [45]. The proline pathway through ProDH is to be added to the list of ET pathways converging at the Q-junction with potential additive effects on OXPHOS and ET capacities when operating in combination.

We found that S was the dominant α-pathway with *J*_S_ > *J*_Pro_. With this understanding, flux control ratios are 𝛼 = 𝐽_S_/𝐽_SPro_ and 𝛽 = 𝐽_Pro_/𝐽_SPro_. Additivity $A_{\overset{ˇ}{\alpha \beta }}$ is defined as (1 − 𝛼)/𝛽 [29]. Complete additivity ($A_{\overset{ˇ}{\alpha \beta }}$ = 1) is obtained when the linear sum of the component S- and Pro-pathway flux (*J*_S_ + *J*_Pro_) equals the flux of the convergent SPro-pathway with the SPro-substrate combination (*J*_SPro_). In the OXPHOS state, $A_{\overset{ˇ}{\alpha \beta }}$ was 2 in liver and 1.6 to 2.1 in kidney mitochondria, thereby indicating excess additivity and the synergistic activation of O_2_ flux. This contrasts with the incomplete additivity of convergent NADH- and succinate-linked pathway flux [29]. LEAK respiration, however, is not linearly responsive to ET capacity, as reflected by the lower additivity observed in liver mitochondria. Surprisingly, the additivity of LEAK respiration was negative in kidney mitochondria, which implies the suppression of LEAK respiration by proline and, consequently, an overly proportional effect on coupling efficiency.

In conclusion, our data show that proline can maintain ATP production by the F_1_F_O_-ATPase when CI is impaired. This information should be added to an already long list involving this amino acid in cellular bioenergetics, osmoregulation, stress protection, apoptosis, and cancer cell metabolism (reviewed in [46])—which are addressed in [5,47,48].

## 4. Materials and Methods

### 4.1. Animals

Mice were of mixed 129 Sv and C57BL/6 background. The animals used in our study were of either sex and between 2 and 6 months of age. Data obtained from liver, kidney, brain, or heart mitochondria of mice of a particular gender or age (2, 4, or 6 months) did not yield any qualitative differences, thus all data were pooled. Mice were housed in a room maintained at 20–22 °C on a 12-h light–dark cycle with food and water available ad libitum. The study was conducted according to the guidelines of the Declaration of Helsinki and were approved by the Animal Care and Use Committee of the Semmelweis University (Egyetemi Állatkísérleti Bizottság, protocol code F16-00177 [A5753-01]; date of approval: May 15, 2017).

### 4.2. Isolation of Mitochondria

Liver, kidney, brain, and heart mitochondria were isolated from mice as described in [36,49]. The protein concentration was determined using the bicinchoninic acid assay and calibrated using bovine serum standards [50] using a Tecan Infinite^®^ 200 PRO series plate reader (Tecan Deutschland GmbH, Crailsheim, Germany).

### 4.3. Determination of Membrane Potential (ΔΨmt) in Issolated Mitochondria

Δ*Ψ*_mt_ of isolated mitochondria (0.5 or 1 mg of mouse liver or kidney mitochondria in two ml buffer medium for normoxic or anoxic experiments, respectively; 0.25 or 0.5 mg of brain mitochondria per two ml of medium (the composition of which is described in [32]); 0.25 mg of mouse heart mitochondria per two ml of medium) was estimated fluorimetrically with safranine O [51] or rhodamine 123 [52] and expressed as arbitrary units or calibrated to millivolts as described in [32], acknowledging the considerations elaborated in [53,54] regarding the inhibition of respiration and unspecific binding of safranine. Fluorescence was recorded using a Hitachi F-7000 spectrofluorimeter (Hitachi High Technologies, Maidenhead, UK) at a 5-Hz acquisition rate, at 495 nm and 585 nm excitation and emission wavelengths, respectively, or the Oroboros O2k (Oroboros Instruments, Innsbruck, Austria) equipped with the O2k-Fluo LED2-Module, or the NextGen-O2k prototype equipped with the O2k-Fluo Smart Module, with optical sensors including an LED (465 nm; <505 nm short-pass excitation filter), a photodiode, and specific optical filters (>560 nm long-pass emission filter) [51]. The experiments were performed at 37 °C.

### 4.4. Mitochondrial Respiration

Oxygen consumption was performed polarographically using an Oxygraph-2k. A half to 1 mg of mouse liver or kidney mitochondria in two ml of buffer medium for normoxic or anoxic experiments was used, respectively; 0.25 mg of brain mitochondria per two ml of medium was used. Mitochondria were suspended in 2 ml incubation medium, the composition of which was identical to that for Δ*Ψ*_mt_ determination. The experiments were performed at 37 °C. The oxygen concentration (µM) and oxygen flux (pmol·s^−1^·mg^−1^; negative time derivative of oxygen concentration, divided by mitochondrial mass per volume and corrected for instrumental background oxygen flux arising from oxygen consumption of the oxygen sensor and back-diffusion into the chamber) were recorded using DatLab software (Versions 5.1.0.20 and 7.4.0.4, Oroboros Instruments, Innsbruck, Austria).

### 4.5. Determination of NADH Autofluorescence in Isolated Mitochondria

The NADH autofluorescence was measured using two different instruments: (1) Hitachi F-7000 fluorescence spectrophotometer (Hitachi High Technologies, Maidenhead, UK) and (2) the NADH-Module of the NextGen-O2k (Oroboros Instruments, Innsbruck, Austria). The NADH measurements were performed in a Hitachi F-7000 fluorescence spectrophotometer at a 5 Hz acquisition rate, using 340 and 435 nm excitation and emission wavelengths, respectively. The NextGen-O2k allows for the simultaneous measurement of oxygen consumption and NADH autofluorescence, incorporating an ultraviolet (UV) LED with an excitation wavelength of 365 nm and an integrated spectrometer that records a wavelength range between 450 and 590 nm. The light intensity of the LED was set to 10 mA. A half a mg of mouse liver, kidney, or brain, or 0.25 mg of mouse heart mitochondria were suspended in 2 ml incubation medium, the composition of which was identical to that for Δ*Ψ*_mt_ determination—as described in [32]. The experiments were performed at 37 °C.

### 4.6. Mitochondrial Q Redox State

The coenzyme Q redox state of isolated mitochondria suspended in a buffer composition as described in [32] was followed amperometically using a three-electrode system with coenzyme Q_2_ (CoQ_2_, 1 µM) as a mediator, and using the Q-Module of the NextGen-O2k (Oroboros Instruments, Innsbruck, Austria). The reference electrode was Ag/AgCl/(3M KCl). The auxiliary electrode was made of platinum and the working electrode was fabricated from glassy carbon. The oxidation peak potential of CoQ_2_ measured by cyclic voltammetry was set to the glassy carbon to measure the oxidation of reduced CoQ_2_. The Q redox state was recorded simultaneously with O_2_ flux and rhodamine 123 fluorescence.

### 4.7. Determination of Proline Dehydrogenase Activity

ProDH activity was determined in alamethicin-treated mitochondria immediately after isolation, as described in [53], with minor modifications. Briefly, the reaction was carried out in a 50 mM phosphate buffer (pH 7.4) with 0.25 mg mitochondria, 10 µg alamethicin, and 1 µM cytochrome c and proline concentrations, as indicated in the legends. After 30 min at 25 °C while shaking (500 rpm), the reaction was stopped by the addition of half the final volume 10 *w/v* % trichloroacetic acid and one-tenth the final volume of freshly made 0.1 M 2-aminobenzaldehyde dissolved in 40 *v/v* % ethanol. After 30 min, the absorbance on 440 nm was read against a parallel blank without substrate. The concentration of P5C was calculated using ε = 2.58 mM^−1^·cm^−1^.

### 4.8. Reagents

Standard laboratory chemicals, duroquinone, tetrahydro-2-furoic acid, and S-5-oxo-2-tetrahydrofurancarboxylic acid were obtained from Sigma Aldrich (St Louis, Missouri, US). SF6847 and atpenin A5 were purchased from Enzo Life Sciences (ELS AG, Lausen, Switzerland). The mitochondrial substrates were dissolved in bi-distilled water and titrated to pH 7.0 with KOH. ADP was purchased as a K^+^ salt of the highest purity available (Merck KGaA, Darmstadt, Germany) and titrated to pH 6.9.

The concentrations of glutamate (G), malate (M), and oxoglutarate (Og) were always 5 mM when present. Succinate was added where indicated (S, 5 mM). ADP concentrations were 2 mM where titrations are indicated. Rotenone (Rot, 1 µM), myxothiazol (Myx, 1 µM; or 0.1 µM when specified), carboxyatractyloside (CAT, 1 µM), oligomycin (Omy, 10 µM), NaCN (CN, 1 mM), or SF6847 (SF, 1 µM; or 0.25 µM when specified) were used as indicated.

### 4.9. Figures on Time Courses

All traces are representative of at least three independent experiments. At the end of many experiments, the uncoupler SF6847 (SF) was added to confer a complete collapse of Δ*Ψ*_mt_ as a point of reference.

### 4.10. Statistics

Statistical analysis was performed by comparing pooled raw data using one-way ANOVA (Bonferroni test) or ANOVA on Ranks if normality failed (details in respective legends); statistical significance was accepted when *p* < 0.05 (shown as *).

## Data Availability

Data available in a publicly accessible repository; The data presented in this study are openly available in https://zenodo.org/deposit/6323145, accessed on 28 March 2022.

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
