# Peer review of "Proline Oxidation Supports Mitochondrial ATP Production When Complex I Is Inhibited"

_ijms, 2022, doi:10.3390/ijms23095111_

Round 1

Reviewer 1 Report

This research article by Pallag et al. dealing with Proline oxidation is quite of interest. Here are my comments about this work :

  • Introduction is too short and not enough emphasized.
  • Results are very difficult to understand and interpret – should definitely be displayed in another way for the readers …
  • Statistic tests are lacking between each of the compared conditions and should definitely be added.
  • Discussion part is short, not enough contrasted in my opinion, and too much « technical ». Does this work fit the scope oh this journal ? I do not know, perhaps it would better to submit in another appropriate journal belonging to MDPI and not IJMS.

Author Response

We thank Reviewer #1 for the comments.

Introduction is too short and not enough emphasized.

> Response: In the revised manuscript, introduction has been slightly altered. Please also note that figure 1 and the entire subsection 2.1 is actually part of the introduction, but as it is a “figure” it is included in the results section. Reference to a figure is not made in the Introduction section.

Results are very difficult to understand and interpret – should definitely be displayed in another way for the readers …

> Response: In the revised manuscript, the results section has been slightly altered. Please note that there are 11 figures in the main text and 13 in the supplement, most comprised of 4-10 panels; the results have been written to the simplest way possible, given the sheer volume of data.

Statistic tests are lacking between each of the compared conditions and should definitely be added.

> Response: In the revised manuscript, statistical analysis has been added in revised figures 2 (A, C-L) and 5 (A-E).

Discussion part is short, not enough contrasted in my opinion, and too much « technical ». Does this work fit the scope oh this journal ? I do not know, perhaps it would better to submit in another appropriate journal belonging to MDPI and not IJMS.

> Response: In the revised manuscript, discussion has been slightly altered and expanded. Regarding the comment whether our work fits in the scope of the journal, the submission is in response to the special issue entitled “Mitochondrial Bioenergetics in Different Pathophysiological Conditions 2.0”. We cannot think of a better subject for showing the effect of mitochondrial bioenergetics conferred by proline, when the pathophysiological conditions involve inhibition of complex I. 

Reviewer 2 Report

I have read carefully this interesting and complex manuscript. Only minor changes are suggested in order to clarify some aspects.

1.- Taken into consideration the complexity of the manuscript and the high level of results provided, a deep discusion is needed especially explaining the results in which rotenone inhibits the effect of proline (Figure 4). As authors indicate that proline can be used in complex I deficiency as fuel to maintain mitochondrial redox potential, why rotenone, that inhibits complex I activity, reduced the effect of proline in assay show in figure 4?

2.- As the use of Oroboros Next-Gen-O2k is relatively new, can the authors confirm the redox status of CoQ by standardized HPLC measurements? This data will confirm the results shown with Oroboros technology.

Author Response

We thank Reviewer #2 for the comments.

1.- Taken into consideration the complexity of the manuscript and the high level of results provided, a deep discusion is needed especially explaining the results in which rotenone inhibits the effect of proline (Figure 4). As authors indicate that proline can be used in complex I deficiency as fuel to maintain mitochondrial redox potential, why rotenone, that inhibits complex I activity, reduced the effect of proline in assay show in figure 4?

> Response: The Reviewer is correct to point out this apparent discrepancy; we offer the following explanation which we also include in the revised manuscript: catabolism of proline leads to P5C and subsequently to glutamate (through GSA), as shown in figure 1; glutamate will be subsequently catabolized in the citric acid cycle, which requires NAD+, mostly coming from oxidation of NADH by complex I. Thus, there are many steps through which proline catabolism is connected to complex I, this is also why we reported in the manuscript that the effect of proline in complex I-inhibited mitochondria on Q redox state was “less pronounced” than in the absence of rotenone (inhibition of complex I would lead to a NADH build-up that would affect upstream catabolism of GSA, in turn affecting P5C formation). However, complex I inhibition is too far downstream from ProDH, thus there is still an effect of proline on Q, but less pronounced. These considerations are included in the results and discussion of the revised manuscript.

2.- As the use of Oroboros Next-Gen-O2k is relatively new, can the authors confirm the redox status of CoQ by standardized HPLC measurements? This data will confirm the results shown with Oroboros technology.

> Response: The Q-Module of the NextGen-O2k has been established since 1988, developed by Peter R Rich (Rich PR (1988) Patent of Q-electrode. Glynn Res. Ph. European Patent no.85900699.1.). As we referenced in our paper citing the methodology (Komlódi T, Cardoso LHD, Doerrier C, Moore AL, Rich PR, Gnaiger E (2021) Coupling and pathway control of coenzyme Q redox state and respiration in isolated mitochondria. Bioenerg Commun 2021.3. https://doi.org/10.26124/bec:2021-0003), Zannoni et al (Measurement of the redox state of the ubiquinone pool in Rhodobacter capsulatus membrane fragments. FEBS Lett 271:123-7.) and Van den Bergen et al (Van den Bergen CW, Wagner AM, Krab K, Moore AL (1994) The relationship between electron flux and the redox poise of the quinone pool in plant mitochondria. Interplay between quinol-oxidizing and quinone-reducing pathways. Eur J Biochem 226:1071-8.) compared the HPLC-based Q-extraction method with the Q-electrode system and revealed that the results of the two methods agree well.

Round 2

Reviewer 1 Report

I would like to thank the authors for sending their revised version of the manuscript. However, authors efforts to respond to all my raised comments are very modest, unfortunately. Introduction is still no clear and too short, Results are still too long (11 Figures ) with a lot of supplementals, thus rendering the work difficult to interpret - authors have indeed partially respond to my comments and did it too quickly ...

Also and most importantly, when submitting a manuscript, authors should avoid self citation - here, 8 papers of the last author (chinopoulos) are referred.

Based on all these comments, I have the regret to announce that I still cannot endorse publication of this work.

Author Response

Dear Dr. Atlante,

thank you very much for your Editorial efforts and for sorting this out.

- Figure 1 has been removed from the “results” section, it is now shown as an explanatory cartoon in the introduction. A simplified version of this cartoon is also shown as a graphical abstract. Figures are thus renumbered and decreased to 10.

-Regarding self-citations:

Ref 18 is required for citing the influence of ATP/ADP ratio on the directionalities of F1FO-ATPase and the adenine nucleotide translocase (ANT). 18. Chinopoulos, C. The “B space” of mitochondrial phosphorylation. J Neurosci Res 2011, 89, 1897-1904, doi:10.1002/jnr.22659.

Ref 32 is required for citing all other parameters influencing the directionality of ANT. 32. Chinopoulos, C.; Gerencser, A.A.; Mandi, M.; Mathe, K.; Torocsik, B.; Doczi, J.; Turiak, L.; Kiss, G.; Konrad, C.; Vajda, S.; et al. Forward operation of adenine nucleotide translocase during F0F1-ATPase reversal: critical role of matrix substrate-level phosphorylation. FASEB J 2010, 24, 2405-2416, doi:10.1096/fj.09-149898.

Ref 33 is required for citing the reversal potential of the F1FO-ATPase. 33. Chinopoulos, C. Mitochondrial consumption of cytosolic ATP: not so fast. FEBS Lett 2011, 585, 1255-1259, doi:10.1016/j.febslet.2011.04.004.

Ref 37 is required for citing the effect of 2-ketobutyrate on mSLP, due to the ATP-consuming propionyl-CoA carboxylase step. 37. Bui, D.; Ravasz, D.; Chinopoulos, C. The Effect of 2-Ketobutyrate on Mitochondrial Substrate-Level Phosphorylation. Neurochem Res 2019, 44, 2301-2306, doi:10.1007/s11064-019-02759-8.

Ref 38 is required for citing the phenomenon that when mitochondria respire on pyruvate or other widely used NADH-linked substrates such glutamate or oxoglutarate, inhibition of the electron transport system at the level of CI leads to F1FO-ATPase reversal. 38. Chinopoulos, C.; Tretter, L.; Adam-Vizi, V. Depolarization of in situ mitochondria due to hydrogen peroxide-induced oxidative stress in nerve terminals: inhibition of alpha-ketoglutarate dehydrogenase. J Neurochem 1999, 73, 220-228, doi:10.1046/j.1471-4159.1999.0730220.x.

Ref 41 is required for citing the CI bypass concept. 41. Ravasz, D.; Kacso, G.; Fodor, V.; Horvath, K.; Adam-Vizi, V.; Chinopoulos, C. Reduction of 2-methoxy-1,4-naphtoquinone by mitochondrially-localized Nqo1 yielding NAD(+) supports substrate-level phosphoryla-tion during respiratory inhibition. Biochim Biophys Acta Bioenerg 2018, 1859, 909-924, doi:10.1016/j.bbabio.2018.05.002.

Ref 45 is required for citing the mitochondrial preparation method. 45. Chinopoulos, C.; Vajda, S.; Csanady, L.; Mandi, M.; Mathe, K.; Adam-Vizi, V. A novel kinetic assay of mitochondrial ATP-ADP exchange rate mediated by the ANT. Biophys J 2009, 96, 2490-2504, doi:10.1016/j.bpj.2008.12.3915.

Ref 50 is required for citing the pitfalls of the safranin calibration method.  50. Chinopoulos, C.; Adam-Vizi, V. Mitochondrial Ca2+ sequestration and precipitation revisited. FEBS J 2010, 277, 3637-3651, doi:10.1111/j.1742-4658.2010.07755.x.

I hope that the re-revised manuscript is suitable for publication.

Sincerely,

Christos Chinopoulos MD, PhD

Round 3

Reviewer 1 Report

I would like again to thank the authors for having made substantial revisions in order to improve quality of the manuscript – It is indeed now an improved version when compared to the previous ones. Here are my last comments before acceptation of this work:

  • Title should be modified and shortened: Proline oxidation supports mitochondrial ATP production when complex I is inhibited.
  • English language should again be improved, for example line 27 “It is concluded that proline …”-> In conclusion, we can state that proline catabolism… Line 343 “when mitochondria respire on pyruvate …” -> When mitochondria depend on pyruvate to support respiration …
  • Line 44, “from various tissues …” which ones? Please add this information.
  • References should be added at the end of sentences and not in the middle, such as those found for example lines 51 and 52. Please do the necessary modifications for all the manuscript to improve clarity.
  • Figure 1, add in the figure the titles describing which organ is displayed -> liver and kidney and not only in the legend – Would clearly improve clarity. Please do the necessary modifications for all the displayed figures when required.
  • Please add this seminal reference in the discussion part: 10.15252/embj.2019103334
  • Similar findings were obtained in this 2018 study in the context of complexes bypassing – please discuss it and definitely add the reference: 10.3389/fneur.2018.00552.
  • In the last paragraph of the discussion part, please also add this seminal reference dealing with cell metabolism in the context of cancer: 10.3390/cancers12051119.

Author Response

We thank Reviewer #1 for the comments.

Title should be modified and shortened: Proline oxidation supports mitochondrial ATP production when complex I is inhibited.

> Response: In the revised manuscript, the title was changed to exactly as suggested by the Reviewer.

English language should again be improved, for example line 27 “It is concluded that proline …”-> In conclusion, we can state that proline catabolism… Line 343 “when mitochondria respire on pyruvate …” -> When mitochondria depend on pyruvate to support respiration …

> Response: In the revised manuscript, the co-author T.N.S. who is a native English speaker, has checked and corrected the manuscript thoroughly.

Line 44, “from various tissues …” which ones? Please add this information.

> Response: In the revised manuscript, the tissue origin (liver, kidney, heart and brain) has been added in the text.

References should be added at the end of sentences and not in the middle, such as those found for example lines 51 and 52. Please do the necessary modifications for all the manuscript to improve clarity.

> Response: In the revised manuscript, unless the sentence involves more than one concept in which references are added at the end of the concept and not at the end of the sentence, references are added at the end, as suggested by the Reviewer.

Figure 1, add in the figure the titles describing which organ is displayed -> liver and kidney and not only in the legend – Would clearly improve clarity. Please do the necessary modifications for all the displayed figures when required.

> Response: In the revised manuscript, in the figure panels we describe the organ, but only in the case where two or more organs are displayed in the same figure.

Please add this seminal reference in the discussion part: 10.15252/embj.2019103334

> Response: In the revised manuscript, this reference has been added; however, mindful that this work refers to proline biosynthesis, not oxidation and that our subsequent, ongoing work (not yet submitted) is addressing the ability of proline to rescue cancer bioenergetics when either complex I is inhibited or glutamine is not available, and that some cells within the same tumor produce proline, while other cells consume it, we would like to “tone-down” the relation of proline to cancer as we wish to address this specifically in our next project.

Similar findings were obtained in this 2018 study in the context of complexes bypassing – please discuss it and definitely add the reference: 10.3389/fneur.2018.00552.

> Response: In the revised manuscript, this reference has been added in section 2.7.

In the last paragraph of the discussion part, please also add this seminal reference dealing with cell metabolism in the context of cancer: 10.3390/cancers12051119.

> Response: In the revised manuscript, this reference has been added and discussed. Please note that our subsequent, ongoing work (not yet submitted) is addressing the ability of proline to rescue cancer bioenergetics when either complex I is inhibited or glutamine is not available. Thus, we would like to “tone-down” the relation of proline to cancer as we wish to address this specifically in our next project.